# Seasonal Testing, Results, and Effect of the Pandemic on Coxsackievirus Serum Studies

**DOI:** 10.3390/microorganisms12020367

**Published:** 2024-02-10

**Authors:** Ramesh Kordi, Arthur J. Chang, Mark D. Hicar

**Affiliations:** 1Department of Pediatric Infectious Diseases, State University of New York at Buffalo, Buffalo, NY 14203, USA; rameshko@buffalo.edu; 2Division of Pediatric Infectious Diseases, University of Nebraska Medical Center, Omaha, NE 68198, USA; archang@unmc.edu

**Keywords:** coxsackievirus A (CVA), coxsackievirus B (CVB), complement fixation, indirect immunofluorescence assay, seroprevalence, SARS-CoV-2 pandemic

## Abstract

Coxsackieviruses (CVs) are common causes of infections and can be life-threatening. Unfortunately, rigorous studies guiding the clinician in interpreting CV serum antibody titer testing is lacking. To explore the epidemiology of circulating CVs and the serological test utility in aiding diagnosis of CV infections in our community, we obtained results of CV immunologic diagnostic tests between 2018 and 2022 from a regional healthcare database. For CV type A, rare individuals had positive CF (complement fixation) tests whereas all 16 individuals with IFA testing showed at least one positive serotype. For CV type B CF testing, 52.2% of 222 patients had at least one serotype positive, with B5 being most common and also the most common with higher titers (14.8% with ≥1:32). We found a significant reduction in seropositivity rate during the pandemic in 2020 compared to 2018, which continued through 2022 (OR: 0.2, 95% CI: 0.08–0.49, *p*-value < 0.001). During the pandemic, the seasonal pattern of positive tests varied from the pre-pandemic pattern. Testing for CVs was increased after the first year of the pandemic. Overall, the variability by month and seasonal change in our data support that CF testing can be used to identify recent CVB infection.

## 1. Introduction

Enteroviruses are a public health concern, given their ubiquitous nature and ability to cause fatal or chronic diseases [1]. A member of the family *Picornaviridae*, genus *Enterovirus* are positive-sense single-stranded non-enveloped RNA viruses. Named for their main transmission route of fecal–oral spread, transmission can also occur via respiratory droplets [2]. There are ten recognized species of *Enterovirus* (EV A–J), with EV species A-D including over 100 different serotypes that can cause diseases in humans [3]. Non-polio enteroviruses are classified into coxsackievirus A (CVA), coxsackievirus B (CVB), echoviruses, and recently identified numbered enteroviruses (such as EV-D68 and EV-A71) [4]. 

Enteroviruses circulate widely across the globe and affect individuals of all age groups [1,5,6,7,8,9,10,11,12,13]. Infection occurs equally in males and females and is more prevalent in summer and fall in temperate regions [2,14]. Non-polio enteroviruses may cause a wide range of clinical presentations, from asymptomatic or mild viral clinical syndromes with or without focal manifestations to life-threatening diseases [15]. While most cases are sporadic, there have been major outbreaks involving multiple serotypes, sometimes resulting in significant morbidity or mortality, reported worldwide [14,16,17,18,19,20,21,22]. Enteroviruses may shed for a prolonged period, but many cases may go unrecognized. 

Herpangina and hand, foot, and mouth disease (HFMD) occur worldwide, predominantly in infants and children. The majority of enteroviral serotypes causing herpangina and HFMD belong to coxsackievirus A [23,24,25,26]. Outbreaks of HFMD have emerged most commonly with coxsackievirus A6 [25,27,28,29,30], A10 [31,32], A16 [33,34], and EV-A71 [34,35]. Although CVA infections are typically mild, there have been rare reports of fatalities associated with coxsackievirus A16 infection, including cases of infantile myocarditis [36] and pneumonitis in an adult [37]. Furthermore, in recent years, an increased number of acute flaccid myelitis cases have been noticed associated with EV-D68, EV-A71 [38,39], and CVA24 [40,41].

Coxsackievirus B can cause severe conditions, including myocarditis, meningitis, encephalitis, hepatitis, and septic-like illness, more frequently in infants and neonates [20,42,43,44,45]. Immunocompromised patients might also be prone to severe life-threatening CVA or CVB infections, especially encephalitis [46,47,48,49]. Significant CVB outbreaks have occurred in various countries, leading to fatal myocarditis in infants [50,51] and meningoencephalitis [52,53]. Studies suggest that CVB can be transmitted vertically, leading to severe neonatal hepatitis, myocarditis, meningoencephalitis, and disseminated intravascular coagulopathy [45,54,55]. CVB is a common pathogen in viral acute myocarditis in infants, children, and adults [56,57,58,59]. Between 10% and 20% of CVB-induced acute myocarditis may progress to dilated cardiomyopathy [59,60,61], which is believed to be attributed to persistent viral non-replicating RNA remnants in myocardial cells and continued autoimmune response [59,62,63,64,65]. Enterovirus genomes were found using reverse transcription-PCR (RT-PCR) in about 35% of explanted heart samples from patients with end-stage dilated cardiomyopathy [66]. Multiple epidemiological studies have also demonstrated a strong association between coxsackievirus B and type 1 diabetes mellitus [67,68,69,70,71]. It has been postulated that persistent CVB RNA remnants may induce cell autoimmunity and islet cell damage [72].

Most reports on enterovirus infections in the United States have been based on passive surveillance conducted by laboratories through the National Enterovirus Surveillance System (NESS), which relies on voluntary participation from laboratories [5,12,13]. The most detected serotypes, as well as their respective frequencies, have changed over time. In 2007–2008, CVB1 was reported for the first time as the most prevalent serotype in the United States, which was associated with clusters of severe neonatal myocarditis, including fatalities [73]. Based on NESS reports, the circulation of enterovirus serotypes has exhibited either epidemic patterns, such as with CVB1, CVB3, and CVB5, characterized by occasional peaks lasting one to three years, or endemic patterns, such as with CVB2, CVB4, and CVB6, with stable and usually low-level circulation [5]. The mechanism behind the cyclic epidemic pattern of enteroviruses has not been fully understood. However, the population herd immunity to a specific serotype, cross-serotype immunity, and virus evolution/mutation have been proposed [74]. Since NESS relies on voluntary reports from participating laboratories, and many cases of CVB infection present with mild or non-specific symptoms and go unrecognized, the reports may not accurately reflect the actual circulation of enteroviruses in the population.

Diagnosis of acute coxsackievirus infection via serology can be used, but the limited literature on the duration and significant cutoffs, potential for cross-reactivity, and requirement for paired acute and convalescent testing make it challenging to determine the usefulness of serological testing. In practice, paired samples are either difficult to obtain or rarely pursued by clinicians.

Current testing using serology also has a number of varying assays. Neutralization antibody assay involves incubating the sample presumably containing antibodies with the target virus. If the sample contains antibodies, they will bind to virus and prevent it from infecting a susceptible cell culture [10]. Neutralization assays are technically complex and typically take 7–12 days to have results. The complement fixation test (CF) is a traditional method to detect the presence of a specific antibody or antigen in a patient’s serum and can return results within 24 h. The patient’s serum is heated in order to be depleted of complements. Then, a known amount of standard complement protein and the antigen of interest are added to the serum. In the next stage, sheep red blood cells (sRBCs) pre-bound with anti-sRBCs are added. If the patient’s serum contains antibodies that fix the complement (IgG1 and IgG3 or IgM), the antigen–antibody complexes consume complement proteins, thereby depleting the complement reaction with sRBC–antibody complexes, and subsequent hemolysis will not occur. The CF test cannot distinguish between IgG and IgM [75]. Indirect immunofluorescent assay (IFA) uses secondary antibodies labeled with a fluorophore that specifically binds to the primary antibodies. The fluorescence emitted by the secondary antibodies will prove the existence of the complex of primary antibody-antigen in the patient’s serum [76]. According to experimental respiratory infections in volunteers and murine studies, the adaptive immune response following coxsackievirus infection is consistent with other viral infections [77,78,79,80]. Coxsackievirus-neutralizing IgM antibodies appear 3 days after infection, reach their peak on day 7, and typically vanish around 3 months after exposure. Anti-CV IgG antibodies released by adaptive B cells appear on day 4 after infection, reach their maximum level 2 to 3 weeks after exposure, and persist for years, although they may decrease gradually over time [81,82]. 

In this study, we investigated the coxsackievirus serum results from a regional healthcare database. We described the seasonal pattering of both testing and results to gain insights into the ability to describe past infection, herd immunity, and overall virus circulation.

## 2. Materials and Methods

HEALTHeLINK, a not-for-profit regional health information organization (RHIO) in Western New York, was used as the data source for antibody tests. This study was approved by the University at Buffalo IRB (# STUDY00006097) and by the HEALTHeLINK Research Committee. The HEALTHeLINK database encompasses data from hospitals in eight counties of Western New York (WNY), including Erie, Niagara, Chautauqua, Cattaraugus, Wyoming, Allegany, Genesee, and Orleans. The HEALTHeLINK database was queried using the SNOMED-CT identifier 117778005 (measurement of human coxsackievirus A antibody) and SNOMED-CT identifier 117787001 (measurement of human coxsackievirus B antibody) from results obtained between 1 January 2018 and 1 January 2023. Results obtained included test specifics, associated LOINC code (Logical Observation Identifiers Names and Codes), associated test code, result, date of test, and age of patient (in years). 

LOINC code results obtained included CVA IFA IgG serology (A7 59584-3, A9 58792-3, A16 59583-5, A24 59582-7); CVA CF serum antibody (A2 9753-5, A4 9754-3, A7 9755-0, A9 9757-6, A10 9750-1, A16 6688-6); CVB CF serum antibody (B1 5104-5, B2 5106-0, B3 5108-6, B4 5110-2, B5 5112-8, B6 5114-4); and CVB semi-quantitative neutralization titers (B1 5103-7, B2 5105-2, B3 5107-8, B4 5109-4, B5 5111-0, B6 5113-6). Per manufacturer’s testing [LabCorp, Burlington, NC, USA], positive result cutoffs were as follows: CVA IFA 1:100, CVA CF 1:8, CVB CF 1:8, and CVB semi-quantitative neutralization 1:10. Data obtained were then analyzed for duplicate entries (as we did not obtain specific patient identifiers, duplicates were defined as two test results with identical serotype, age of individual, result date, and titer). For CVB CF testing, 129 duplicates were removed from the original 881 test results leaving 752 for analysis. No other modality had duplicates noted. To investigate the impact of the SARS-CoV-2 pandemic on testing and seropositivity rates, we separated the data into pre-pandemic (2018 through 2019) and during the pandemic (2020 through 2022). 

We analyzed the data using SPSS 29 software (SPSS Inc., Chicago, IL, USA). Descriptive analysis and proportions were calculated to determine seroprevalence in different categories, and the chi-square test was used to compare seropositivity rates between groups. Logistic regression was performed to assess the association between seropositivity rate and age as well as year. We considered *p*-values less than 0.05 as statistically significant. Figures were created using Excel (Microsoft Corporation, Redmond, WA, USA; version 16).

## 3. Results

### 3.1. Coxsackievirus B

There were no IFA and only 15 CVB neutralization assay results from a restricted time period in the dataset. Five of the 15 neutralization assay results were positive (>1:10): B2 of 1:40 and 1:80, B4 of 1:320 and ≥1:640, and B5 1:80. 

We obtained data from a total of 1637 CF tests for both CVA (see Section 3.2) and CVB. The CVB CF dataset revealed a robust dataset, with 752 tests from 222 patients with a median age of 42 (range: 3–80). The median number of serotype results obtained (B1–B6) per individual was three. A total of 266 tests (35.3%) were positive with titers ≥1:8 (Figure 1a), and 40 CVB CF tests ≥1:32 were seen in 29 individuals (Figure 1a,b). From 222 patients, 116 (52.2%) had one or more positive serotypes (highest titer achieved shown in Figure 1b). 

The most common positive serotype was the B5 serotype (62 patients), followed by B1 (54 patients) and B6 (51 patients). The B4 serotype had the lowest seropositivity (21 patients) (Figure 2). The serological titers ≥1:32 for each CVB serotype were as follows: B1 (5/132, 3.7%), B2 (6/114, 5.2%), B3 (4/126, 3.1%), B4 (1/134, 0.7%), B5 (19/128, 14.8%), and B6 (5/118, 4.2%). 

Given that the data were obtained from individuals with a wide age range, we categorized testing by age group. Among 266 positive tests, 171 (64.2%) were found in individuals between the ages of 30 and 60. Beyond that, there was no discernable pattern of positive results in decadal adult groups (ages 20–29, 30–39, 40–49, 50–59, or 60+); total positive (titers ≥1:8) were 34–65 tests per group and high titer (titers ≥1:32) was 7–9 per group. We did not observe any significant variation in the rate of seropositivity when comparing individuals aged 21 years or younger to those older than 21 years, using a positive cutoff point of ≥1:8 (29/103, 28.1% vs. 237/649, 36.5%, *p* = 0.09), although this analysis reveals that the database accessed may favor data obtained from adults. 

In analyzing the seasonality of the data, it is evident that positive tests were more frequently detected during the summer and fall seasons (Figure 3). The data exhibits a dual-peak distribution, with the lowest point observed in May. The majority of serum titers, specifically those ≥1:32, were observed between July and October (29 out of 40, 72.5%). 

To assess the effect of the pandemic on the seasonal pattern of testing results, we compared the presence of CVB antibodies between the periods of 2018–2019 and 2020–2021. There was a marked shift in the monthly distribution of positive cases and the positivity rate (Figure 4) away from summer months during the pandemic.

We observed a decrease in the number of positive tests and the positivity rate in 2020, when SARS-CoV-2 was first seen in our area. The positivity rate increased relatively in 2021, then declined, despite an overall increase in testing (Figure 5 and Table 1). The number of individuals with at least one positive serotype and the seropositivity rate decreased in 2020 (Figure 6). We found a significantly higher rate of test positivity in the pre-pandemic period (2018–2019) compared to the pandemic period of 2020–2022 (141/295, 47.8% vs. 125/457, 27.4%; *p* < 0.001). This variance seemed to be driven by an overall increase in testing for CVs during the pandemic.

Among individuals, CVB seropositivity rate was almost equal among various age groups of 0–20, 21–40, and 41–60. Individuals older than 60 had relatively a lower seropositivity rate than other age groups; however, the difference was not statistically significant. The seropositivity rate decreased significantly in 2020 as compared to 2018 and continued to decline through 2022 (Table 2).

### 3.2. Coxsackievirus A

In contrast to the robust dataset obtained for CVB, there were more limited data for CVA. A total of 885 CVA CF test results from 257 patients (median age 42, range 6–89) was reviewed. The total numbers of tests conducted were 132 in 2018 and 182 in 2019. Tests experienced a decline in 2020 (134 tests), correlating with the beginning of the SARS-CoV-2 pandemic. The highest two annual test totals were in 2021 (196) and 2022 (241), similar to the pattern of testing seen for CVB. Only six patients (2.3%) had a positive test result. Notably, there were no positive results observed for the A7 and A10 serotypes. Notably, no positive CVA serological tests were detected using CF in 2021 and 2022.

A total of 35 CVA IgG IFA results (positive at a cutoff of ≥1:100 for serotypes A7, A9, A16, and A24) was obtained, and no IgM results were available. These tests were from a total of 18 patients (median age 51; ages: 22–73). Positive results showed no discernable pattern in age or year of study. Three of the 18 individuals were found to have a positive titer for only a single strain (2 A24 and 1 A9) with only 1:200 as the highest titer. Of the remaining 15 with multiple positive serotypes, titers were identical across the tested serotypes in 12 cases. 

## 4. Discussion

This study shows that, similar to other viruses, amount of testing and CV seasonal patterning was affected by the SARS-CoV-2 pandemic. The variance in the month-to-month positivity of CVB CF results supports the transient nature of CF testing. As CF testing reflects predominantly IGM complement fixation, and the pattern of positive testing shown here parallels known epidemiologic patterns of coxsackievirus circulation, this supports the idea that CF testing mostly reflects recent infection. Ideally, using nucleic acid testing during acute presentation or comparison of acute and convalescent titers would be used for diagnosis. In lieu of such testing, our current study suggests that CVB CF positive testing can support the diagnosis of recent CVB infection in clinically appropriate individuals. 

This study does not have enough data to support CF testing for CVA, however. Coxsackieviruses cause a wide range of clinical diseases from mild non-specific viral infections to serious conditions including severe pulmonary infections, myocarditis, and meningo-encephalitis. HFMD can be caused by various enterovirus serotypes, including EV-A71, A6, A10, and A16 [33,83,84,85,86,87]. A seroprevalence study on EV-A71 and CVA6 was performed in the United Kingdom from 2006 to 2017 including 1573 residual samples. Using a microneutralization assay, the seroprevalences of EV-A71 and CVA6 were 32% and 54%, respectively, in the 6–11 month age group, and it increased to over 75% by the age of 10 [88]. A meta-analysis of 71 studies from 13 countries reported that the seroprevalence of EV-A71 ranged from 4.31% to 88.8% and CVA6 from 40.8% to 80.9% [89]. We had very few positive CVA serotypes via CF (2.6%), possibly due to few children being included in this study or a more limited CF CVA response during infections. Therefore, it is difficult to draw firm conclusions related to CVA testing through CF. To the best of our knowledge, there is no sero-epidemiological study on CVA using the CF method. 

Our CVA CF results are in contrast to CVA IFA testing, where all of the 18 individuals tested were positive. The high sensitivity and specificity (>95%) of the IFA kit have been shown to be useful for detection of CVA serotypes [90]. However, as the IgG antibodies’ assay with IFA can persist for years, likely these reflect remote past infections. Moreover, among 12/15 (80%) individuals having multiple CVA IFA tests, the titers were identical across tested serotype, which suggests antibody cross-reactivity between serotypes through this method. It is unclear why hospitals in our region have not utilized IFA IgM testing assays. 

Although most CVB infections are mild and non-specific and do not lead to testing, in certain situations such as suspected involvement of the central nervous system, myopericarditis, neonatal infections, or immunocompromised patients, laboratory diagnosis can have implications in management. Commercial molecular tests are sensitive in detecting enteroviral RNA, but they cannot determine the specific serotype as they rely on primers from the conserved 5’-noncoding region of the enterovirus genome [91]. Additionally, commercial enterovirus RT-PCR cannot distinguish between enteroviruses and rhinoviruses due to similarity in the 5’-noncoding region [92]. Serology tests may be considered when a specific serotype is suspected. We found a CVB CF seropositivity rate of 52.2% from individuals presumed to have a viral illness, consistent with other studies globally, indicating the widespread circulation of coxsackievirus B [10,93,94,95]. The positive results could be attributed to acute infection or history of previous exposure. Previous studies have shown a significant percentage of healthy individuals had positive serological CVB antibody results through complement fixation [96], ELISA [95], or neutralizing antibody assay [10,93]. Although setting a higher cutoff point increases the specificity for detection of the acute infection, it reduces the sensitivity [96]. Overall, ideal testing using serological diagnosis of an acute enteroviral infection requires paired acute and convalescent tests separated by at least 4 weeks with a four-fold or greater increase in titer [96,97,98]. Our most common CVB strain was B5, with a paucity of B4 positive tests. In regards to CVB6, we observed a seropositivity rate of 51/118 (43.2%), which is notably different from rare reports of CVB6 using NESS surveillance data [12,13]. This discrepancy can be explained by mild manifestations caused by CVB6, rendering most cases undetected. 

We observed a significant decline in the number of CVB positive results in 2020. This mirrors the decrease observed in viral respiratory illnesses following the SARS-CoV-2 pandemic in most countries [99,100,101]. The implementation of rigorous public health measures such as social isolation, physical distancing, and widespread use of masks and disinfectants can account for the reduction in all viral infections. The CVB seropositivity in our population was reduced in 2020. The lower rate of positivity persisted through 2022 (Table 2); however, this also reflected a general increase in testing (Figure 6). 

There is limited population-based serological data available on CVB. A retrospective study conducted in an academic center in Rome, Italy between 2004 and 2016 examined CVB seropositivity in 2459 subjects [94]. Using a neutralizing antibody assay with a cutoff ≥1:32, 69.1% of individuals were positive for at least one serotype. Seropositivity rates increased with age, from 25.2% in those under 2 years old to 50.3% in the 3–5-year-old group, and 70% in children aged 6–10. The number of positive serotypes increased with age, with a 22.4% positive rate for three serotypes in individuals aged 41 to 50. The study observed significant changes in seropositivity rates over the years, particularly for B1–B5. In our series, we did not find a significant difference in seropositivity rates among different age groups. However, it should be noted that our series had a small number of cases under 10 years old (10/218, 4.6%), which limits the reliability of the results for this age group. In a case-control study conducted between 2001 and 2005 in five European countries (Finland, England, France, Sweden, and Ireland), the prevalence of CVB antibodies was compared between 249 children newly diagnosed with type I diabetes and 249 control children who were matched in terms of sex, age, time of sampling, and country [93]. The children, with a median age of 9 (ranging from 1 to 23), were tested for seropositivity to at least one serotype using the plaque neutralizing assay method. The results showed that seropositivity rates ranged from 28% in children under the age of 4 to over 80% in children older than 8 years. CVB1 and CVB6 were the least prevalent serotypes, while CVB1 showed a significantly higher positivity rate in patients with type I diabetes as compared to the control group. Notably, the prevalence of CVB antibodies was lower in Finland compared to the other countries. In a study conducted in Germany [96], the threshold for detecting anti-enterovirus antibodies was determined using the complement fixation test. They analyzed the results from 200 healthy individuals, which included 50 healthy pregnant women, 50 healthy children, and 100 healthy blood donors. The study revealed that 67% of the healthy individuals had titers equal to or less than 1:32, 24% had titers of 1:64, 6% had titers of 1:128, and 3% had titers greater than 1:256. Considering titers greater than 1:256 as positive, the complement fixation test yielded positive results in 15/43 (21%) of patients diagnosed with enterovirus meningitis confirmed through RT-PCR analysis of cerebrospinal fluid (CSF). This cutoff proved a high specificity of 85% in 105 patients with enterovirus negative meningitis. Similarly, out of 11 patients who were enteroviral RT-PCR positive in stool specimens, titers of 1:256 and ≥1:128 were found in one (10%) and five patients (45.4%), respectively. Choosing a lower cutoff at 1:32 had a higher sensitivity in patients with CSF RT-PCR-positive enteroviral meningitis (31/43, 72%) and in patients with the virus detected in stool using RT-PCR (9/11, 81.8%); however, this cutoff comes at the cost of low specificity. The authors concluded that for diagnosis of acute infection via complement fixation test, paired acute and convalescent tests with at least four-fold increase in antibody titer are required. Positive molecular testing for RNA detection in a relevant sample is the current standard of care for confirming acute infection [97].

Our study has a number of limitations. This study captured data from hundreds of subjects, but lacked clinical data, other testing (including molecular), and International Classification of Disease (ICD) diagnostic data to attempt to clinically correlate these findings. As noted, paired samples with a rise in convalescence are a classic way to confirm an infection, but we did not have the ability to pair these results. As we lacked clinical correlates and paired sampling, we were unable to precisely distinguish acute infections from history of previous infection based on a single positive serological result. As discussed above, the low sensitivity and/or low specificity of serological tests may make the interpretation of the results challenging. This database also lacked results from other available testing, notably IFA IgM-based assays. Our SNOMED query does capture these, so this type of assay is likely not used readily in our region. This database was also an active enrollment dataset, requiring subject written participation, so there is selection bias regarding data available in this database. This was particularly notable in the low number of children included in this database. The data are from patients referred to clinics or hospitals with a presumed presentation that would necessitate testing, so our findings cannot be presumed to be true population seroprevalence. As enteroviruses can vary yearly, five years of data is a small sample size for comparing seasonal patterning, particularly for serotypes with epidemic patterns of circulation such as CVB1, CVB3, and CVB5 [5]. 

There is no study on sero-epidemiology of coxsackieviruses in the United States. Future studies should be aimed at correlating serum testing, clinical presentation, and acute viral infection to further support the use of serum testing for diagnosis. True seroprevalence testing and long-term surveillance (5–10 years) to properly assess the circulation of these serotypes in a community should be pursued. 

## Figures and Tables

**Figure 1 microorganisms-12-00367-f001:**
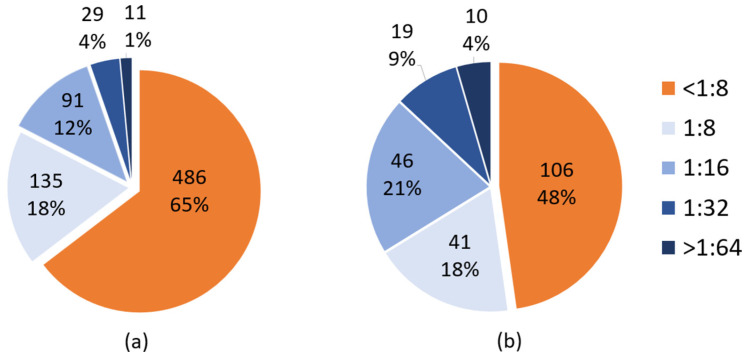
CVB complement fixation (CF) by total test (**a**) and by subject (**b**). (**a**) The orange color indicates the test results on a titer of <1:8. The different blue shades show the number and rate of positive tests for each titer. (**b**) The orange color indicates the number of individuals with negative tests based on a titer of <1:8. The different blue shades show the number of individuals with positive tests. For each individual, the highest CVB titer achieved in any serotype (B1–B6) was considered.

**Figure 2 microorganisms-12-00367-f002:**
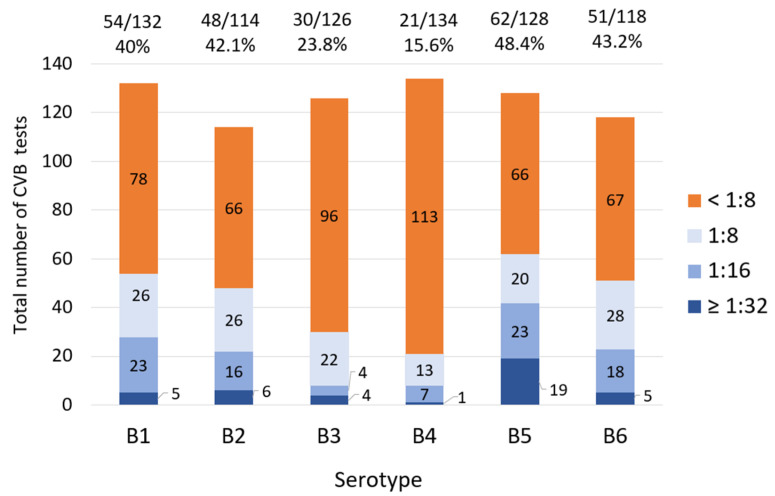
CVB CF titers in each serotype. Each colored bar demonstrates the number of specific titers. The orange bars show the number of negative tests, and different blue shades reveal the number of positive tests with various titers (cutoff ≥1:8). The floating numbers above each bar show the number of positive CVB CF tests/total tests for each serotype.

**Figure 3 microorganisms-12-00367-f003:**
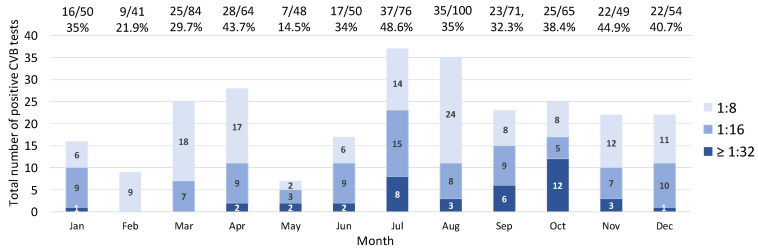
The CVB CF positivity rates and positive titers in each month. The distribution of CVB CF positive tests (cutoff ≥1:8) and seropositivity rates across months in the Western New York community from 2018 to 2022. Each colored bar demonstrates the number of positive tests for various titers. The darkest blue bar shows a titer of 1:32, and the lightest indicates titer of 1:8. As shown, the highest numbers of positive tests are in July and October. The floating numbers above each bar show positive CVB CF tests/total tests in each month.

**Figure 4 microorganisms-12-00367-f004:**
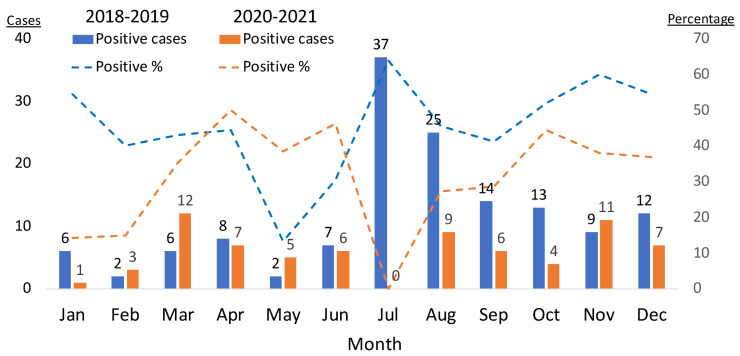
Positive CVB CF tests for each month during 2018–2019 vs. 2020–2021. CVB CF positive tests (cutoff ≥1:8) across months for the year span listed. Cases are noted on left axis and numbered above the bar; percentage positive is noted on right axis. Blue bars represent positive cases; blue dashed line represents rate of positive test percentage for the pre-pandemic period (2018–2019). Orange bars represent positive cases; orange dashed line represents rate of positive test percentage for the period during the pandemic (2020–2021). The number of positive cases decreased notably in the summer during the pandemic.

**Figure 5 microorganisms-12-00367-f005:**
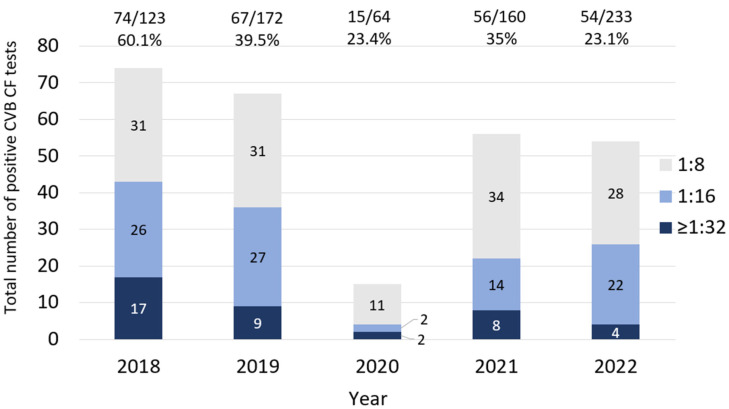
The CVB CF positivity rate and the titers of positive tests in each year. The number of CVB CF positive tests and seropositivity rates (cutoff ≥1:8). Each colored bar demonstrates the number of positive tests with various titers. The floating numbers above each bar show the total positive CVB CF tests/total tests for each year.

**Figure 6 microorganisms-12-00367-f006:**
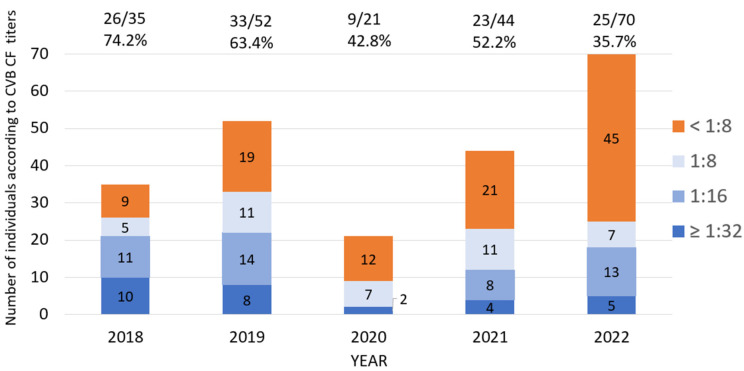
The number of individuals according to the highest level of CVB CF titers in each year. The number of individuals according to the highest level of CVB titers in the tested serotypes from 2018 to 2022. The orange bars show the number of individuals with negative CVB CF in all tested serotypes (cutoff <1:8), and blue shades reveal the number of individuals with various titers of positive tests.

**Table 1 microorganisms-12-00367-t001:** The number of positive serological CVB tests (cutoff ≥1:8) for each serotype in the Western New York community across the years 2018 through 2022.

Serotype	Year/Total Tests					
	2018 123 *	2019 172	2020 64	2021 160	2022 233	Total 752
**B1**	16	9	3	16	10	54
**B2**	12	13	3	10	10	48
**B3**	10	8	1	3	8	30
**B4**	7	8	0	3	3	21
**B5**	16	14	3	14	15	62
**B6**	13	15	5	10	8	51
**Total positive** **% positive**	74 (60.1%)	67 (39%)	15 (23.4%)	56 (35%)	54 (23.1%)	266 (35.3%)

* Total number of CVB CF tests each year.

**Table 2 microorganisms-12-00367-t002:** Coxsackievirus B seropositivity (cutoff ≥1:8) for at least one serotype in individuals according to age and year.

**Factor**	**Positive/Total** ***n* (%)**	**OR ***	**95% CI ****	** *p* ** **-Value *****
**Age:**				
**0–20**	15/27 (55.6)	1	-	-
**21–40**	48/85 (56.5)	0.93	0.43–2.48	0.93
**41–60**	44/80 (55)	0.97	0.40–2.35	0.96
**60+**	10/30 (35.5)	0.4	0.13–1.17	0.09
**Year:**				
**2018**	26/35 (74.2)	1	-	-
**2019**	33/52 (63.5)	0.55	0.21–1.42	0.29
**2020**	9/21 (42.9)	0.26	0.08–0.82	0.02
**2021**	23/44 (52.2)	0.38	0.14–0.95	0.04
**2022**	25/70 (35.7)	0.19	0.07–0.47	<0.001

* odds ratio, ** confidence interval, *** logistic regression *p*-value.

## Data Availability

Data are available by request through the authors.

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
