# Peer review of "Seasonal Testing, Results, and Effect of the Pandemic on Coxsackievirus Serum Studies"

_microorganisms, 2024, doi:10.3390/microorganisms12020367_

Round 1

Reviewer 1 Report

Comments and Suggestions for Authors

In this article, Kordi R., et al., reported a serological survey about the infections of coxsackieviruses during Jan. 2018 - Jan. 2023. Total 885 samples were included in the analysis of Coxsackievirus A (CVA) infection, 752 samples were included in the analysis of Coxsackievirus B (CVB) infection (126 were excluded due to possible duplication). For CVA, rare cases were positive with compliment fixation assay (CFA), and 16 cases were positive with indirect immunofluorescence assay (IFA).  For CVB, 52.2% (222) were positive by CFA. Type B5 was the dominant serotype and with highest titer. Further study showed that there was a significant decline of CVB seropositivity in 2020, suggesting a connection to the disease control of SARS-CoV-2 pandemic. Based on the survey, the authors concluded that CFA can be used to identify recent CVB infection. Overall, the finding is interesting about the CVB infection among Western NY, given that many healthcare workers believe that CVB infection keeps a very low rate. However, the study is not well organized. A vigorous improvement is needed for further consideration.

1. A major concern is that all data are from one serological assay, mostly, by CFA. There is no validation with virus isolation or RNA detection, at least, in part of the samples. The sensitivity and specificity may be a problem for the data interpretation, though there is suggested one provided by the kit manufacturer.      

2. The conclusion is not properly stated. There is no evidence to show that CFA with a single test can be used to identify when the infection occurs, no matter what the titer is. In this study, the rate of CVB positivity in 2020 (23.4%) is similar to that of 2022 (23.1%). The "decline" of 2020 in figure 10 is due to the relatively small sample size. The conclusion should be rephrased.

3. Figure 3b is confused. The legends are not corresponding to the figure. 

4. The duplicated samples based on CVB tests should also be excluded from the analysis of CVA. The total is 752 cases for both CVA and CVB.

5. The study was mainly focused on CVB tests, the CVA detection can be ignored (only mentioned in the DISCUSSION section), or put in the end of the RESULTS section.  

6. About Table 2, it looks like the age groups were listed in the table disorderly? 

7. Some figures are duplicated and can be removed. The figure number is better to reduce to 5 - 6. The figures and tables for CVA are actually not necessary. 

8. Figure 10 has not been cited in the context.  

Author Response

Response to Reviewer 1 comments regarding manuscript entitled “Seasonal testing, Results, and Effect of Pandemic on Coxsackievirus Serum Studies” Submitted to Microorganisms Journal. 

We would like to express our gratitude for the time you take to review our manuscript. Please find our detailed response to each comment below. We track highlighted the changes in the revised manuscript.

Comment 1: A major concern is that all data are from one serological assay, mostly, by CFA. There is no validation with virus isolation or RNA detection, at least, in part of the samples. The sensitivity and specificity may be a problem for the data interpretation, though there is suggested one provided by the kit manufacturer.

Response to comment 1: Thank you for the comment. We mentioned on page 2 lines 86-90, and page 10 lines 359-361that single serological testing has challenges in diagnosis of acute infection and the paired acute and convalescence testing is required for confirming the diagnosis. We also mentioned in the limitation of the study, line 364, that we did not have access to molecular testing. We added in the discussion that molecular testing is the standard of care for detection of acute infection diagnosis (lines 362-363).

We discussed on page 10, lines 345-361, using the results of a study (reference number 96) that complement fixation serological test for detection of CVB acute infection lacks a desired specificity and or sensitivity. We added this point in the limitation of our study as well (lines 369-370).

Comment 2: The conclusion is not properly stated. There is no evidence to show that CFA with a single test can be used to identify when the infection occurs, no matter what the titer is. In this study, the rate of CVB positivity in 2020 (23.4%) is similar to 12/27/23, that of 2022 (23.1%). The "decline" of 2020 in figure 10 is due to the relatively small sample size. The conclusion should be rephrased.

Response to comment 2: We agree the way we phrased the conclusion caused confusion. We have based this conclusion on the seasonal pattern of testing, which is reflected in new figure 3 and 4 – old figures 7-9. Old Figure 10 (new figure 5) rates being lower does reflect that there was an increase in testing during the pandemic. This was further clarified in the text for that figure.

As for the overall conclusion, year to year comparison was not used, but we did look at seasonality or month to month variance. If background positivity rates/cases were the same over every month, this would suggest a longer-lasting immune response. As the CF testing reflects predominantly IGM complement fixation and the pattern of positive testing parallels known epidemiologic patterns of coxsackievirus circulation, we feel these can support a diagnosis in patients that have suggestive symptoms.  We do note later in the discussion that ideally, this would be used with acute and convalescent testing. We have moved up language regarding acute and convalescent testing, nucleic acid testing, and added language to temper our conclusion accordingly.   

Comment 3: Figure 3b is confused. The legends are not corresponding to the figure.

Response to comment 3: Each individual might have multiple positive B1-B6 serotypes. For designing figure 3b, we considered the highest titer achieved in any serotype. We rephrased the legend for figure 3b and added clarifying language to the text.

Comment 4: The duplicated samples based on CVB tests should also be excluded from the analysis of CVA. The total is 752 cases for both CVA and CVB.

Response to comment 4: We had a total 1637 complement fixation tests for CVA (885 tests) and CVB (752 tests) and added this notation to section 3.1 lines 155-156 and section 3.2 line 249.

Comment 5: The study was mainly focused on CVB tests, the CVA detection can be ignored (only mentioned in the DISCUSSION section), or put in the end of the RESULTS section.

Response to comment 5:

Since our methodic approach was to collect that data and that data would be of broader interest, we retained this in the results. Per suggestions, we moved this section to the end, shortened the section and removed figures.

Comment 6: About Table 2, it looks like the age groups were listed in the table disorderly?

Response to comment 6: This table was removed and findings summarized in the text in response to other suggestions.

Comment 7: Some figures are duplicated and can be removed. The figure number is better to reduce to 5 - 6. The figures and tables for CVA are actually not necessary.

Response to comment 7:

Thank you, we have removed the figures and tables for CVA and summarized key findings in the text. We have also removed the figures dealing with age analysis as that data can be summarized in the text. With the edits done to these sections, this has significantly reduced the figures.

To summarize, we have removed figures 1, 2, 5, 6, 12 and tables 1 and 2 and have combined old figures 8 and 9 into new figure 4. Other figures and tables have been numbered accordingly.

Comment 8: Figure 10 has not been cited in the context.

Response to comment 8: With removal of other figures, this is now Figure 6.  Figure 6 (previously Figure 10) is cited on page 6 line 214.

Reviewer 2 Report

Comments and Suggestions for Authors

Kordi et al's manuscript presents the results of a retrospective study on coxsackievirus (type A and type B) testing cases in eight counties of Western New York from 2018 to 2022. The authors examined the epidemiology of circulating coxsackieviruses and evaluated the effect of the coronavirus pandemic on the detection outcomes of coxsackieviruses in the region. The subject matter and results of the manuscript are entirely consistent with the Special Issue "Coxsackievirus Infection and Associated Diseases 2.0" of the Microrganisms journal. The research topic is relevant in terms of coxsackievirus epidemiology and because Coxsackievirus type B can trigger type 1 diabetes.

All sections of the manuscript are well-written, and the results are illustrated with 12 figures and 4 tables. The authors have discussed the results and limitations of their study in detail. I have only minor comments and suggestions.

 Abstract

The abstract presents the primary findings and conclusions of the manuscript.

According to the journal requirements, the abstract should be a maximum of 200 words. The current revision consists of 207 words.

Line 15. Please decipher the CF (Complement fixation) test in the abstract.

 Introduction

This section is well-written and supported by a significant number of references. The authors have provided strong evidence of the clinical relevance of detecting coxsackieviruses of different types (A and B).

Minor issues:

Line 45. Please, correct HDMD to HFMD

Line 68. Authors may cite Carré et al, 2023, Endocr Rev., doi: 10.1210/endrev/bnad007.

Lines 91-113. This paragraph includes information from textbooks that may be redundant, although it can still be useful.

Materials and Methods

This section is correctly described. No issues.

Results

The 'Results' section consists of two sections according to the object of study (Coxsackievirus A and Coxsackievirus B) and includes 4 tables and 12 (!) figures.

Minor issues:

Table 2. It may be useful to introduce vertical delimiters starting with rows 3 and 4 for better clarity.

Line 202, Figure 4 caption. Please, replace 'spicific' with ‘specific’

Line 212, Figure 5 caption. Same as above

There is no reference to Figure 8 in the text (should be in Line 239?)

Line 240. Figure 9 instead of Figure 10?

Discussion

This section is also well written. The authors have thoroughly and comprehensively discussed the major limitations of their study. First, they discussed insufficient data on CF testing for CVA (lines 299-312). In the next paragraph, the authors address the discrepancy between the results obtained from CVA testing using the CF system and ELISA. The authors discuss why they rely on serologic test results instead of molecular (NAAT-based) tests.

The authors conducted a study of Coxsackievirus B cases in eight counties of Western New York. This raises the question of population-based studies and extrapolation of the results to other states in the US. Lines 349-384 provide the answer to this question for EU countries, but not for the US (no data?).

Minor issue:

Please start a new paragraph on line 386 ' Our study has a number of limitations'.

Line 395. Database?

References

The reference list should be reformatted according to the journal's requirements

https://www.mdpi.com/journal/microorganisms/instructions#manuscript

Journal Articles:

1. Author 1, A.B.; Author 2, C.D. Title of the article. Abbreviated Journal Name Year, Volume, page range.

Comments on the Quality of English Language

The English language is proficient. Only a few errors require correction.

Author Response

Response to Reviewer 2 comments regarding manuscript entitled “Seasonal testing, Results, and Effect of Pandemic on Coxsackievirus Serum Studies” Submitted to Microorganisms Journal. 

We would like to express our gratitude for the time you take to review our manuscript. Please find our detailed response to each comment below. We track highlighted the changes in the revised manuscript.

Comment 1: Abstract section.

The abstract presents the primary findings and conclusions of the manuscript.

According to the journal requirements, the abstract should be a maximum of 200 words. The current revision consists of 207 words.

Line 15. Please decipher the CF (Complement fixation) test in the abstract.

Response to comment 1: We observed the comment in the manuscript. The abstract has 200 words now. CF was modified to complement fixation in the abstract as well.

Comment 2: Introduction section.

This section is well-written and supported by a significant number of references. The authors have provided strong evidence of the clinical relevance of detecting coxsackieviruses of different types (A and B).

Minor issues:

Line 45. Please, correct HDMD to HFMD

Line 68. Authors may cite Carré et al, 2023, Endocr Rev., doi: 10.1210/endrev/bnad007.

Lines 91-113. This paragraph includes information from textbooks that may be redundant, although it can still be useful.

Response to comment 2: Thank you for the comments.

Line 45: HDMD was corrected to HFMD.

Line 68: We cited Carré et al, 2023, Endocr Rev., doi: 10.1210/endrev/bnad007 as ref 71.

Line 91-113. We mentioned the basics of various methods of serological testing which were discussed in this manuscript. We assume that it is still useful as a brief review.

Comment 3: Results section.

The 'Results' section consists of two sections according to

the object of study (Coxsackievirus A and Coxsackievirus B)

and includes 4 tables and 12 (!) figures.

Minor issues:

Table 2. It may be useful to introduce vertical delimiters.

starting with rows 3 and 4 for better clarity.

Line 202, Figure 4 caption. Please, replace 'spicific' with

‘specific’

Line 212, Figure 5 caption. Same as above

There is no reference to Figure 8 in the text (should be in

Line 239?)

Line 240. Figure 9 instead of Figure 10?

Response to comment 3:

Table 2: This has been removed in response to Reviewer 1 comments.

Line 202: The spelling was corrected (line  173 in the new version).

Line 212:  Figure 5 was removed.

Figure 8 (now Figure 4) was cited in line 201. The new version of Figure 4 is a combination of old Figures 8 and 9.

Line 240: New figure 4 contains the information of old figures 8 and 9. With removal of some figures, old Figure 10 changed to Figure 5 which is cited in line 212.

Comment 4: Discussion section.

This section is also well written. The authors have thoroughly and comprehensively discussed the major limitations of their study. First, they discussed insufficient data on CF testing for CVA (lines 299-312). In the next paragraph, the authors address the discrepancy between the results obtained from CVA testing using the CF system and ELISA. The authors discuss why they rely on serologic test results instead of molecular (NAAT-based) tests.

The authors conducted a study of Coxsackievirus B cases in eight counties of Western New York. This raises the question of population-based studies and extrapolation of the results to other states in the US. Lines 349-384 provide the answer to this question for EU countries, but not for the US (no data?).

Minor issue:

Please start a new paragraph on line 386 ' Our study has a number of limitations'.

Line 395. Database?

Response to comment 4: Thank you for the comments.

We agree improved epidemiologic studies would be helpful. We mentioned in line 381 that there is no sero-epidemiological study on coxsackieviruses in the United States.

The limitation of the study was started in a new paragraph (line 363 in the revised version).

Line 395: Database was corrected to dataset (line 374 in the revised version).

Comment 5:

References: The reference list should be reformatted according to the journal's requirements https://www.mdpi.com/journal/microorganisms/instructions#m manuscript

Response to comment 5: We downloaded the MDPI.ens file for endnote and changed the citation style.

Round 2

Reviewer 1 Report

Comments and Suggestions for Authors

The revised manuscript has well addressed the questions raised in the previous version. A few terms and words are still in wrong form:

1. Line 29, the family "Picorniviridae" and genus "Enterovirus" should be italic.

2. Line 32, "Enteroviruses" is confused. Use "Enterovirus" (italic form) for the genus or use "enteroviruses" for the viruses. 

3. Line 34 and 39, "nonpolio" or "non-polio", be consistent please.

4. Line 35,  48, 276, 278, "EV-71" and "EV71" should be corrected as "EV-A71".

5. Line 52, "CV-A24" is better to be "CVA24".

6. Line 125, "coxsackie" or "coxsackievirus"?  

Author Response

Thank you once again for the thorough review. All of the issues have been corrected, as noted in red font in the uploaded PDF draft. Also, we adjusted the layout of Fig 6 legend, and Figure 3. 

We provided a clean draft as well. 
